# A quantitative hypermorphic *CNGC* allele confers ectopic calcium flux and impairs cellular development

David M Chiasson[1†], Kristina Haage[1†], Katharina Sollweck[1], Andreas Brachmann[1], Petra Dietrich[2], Martin Parniske[1*]

[1]Faculty of Biology, Institute of Genetics, Ludwig Maximilian University of Munich, Munich, Germany; [2]Molecular Plant Physiology, Department of Biology, University of Erlangen-Nürnberg, Erlangen, Germany

**Abstract** The coordinated control of $Ca^{2+}$ signaling is essential for development in eukaryotes. Cyclic nucleotide-gated channel (CNGC) family members mediate $Ca^{2+}$ influx from cellular stores in plants (Charpentier et al., 2016; Gao et al., 2016; Frietsch et al., 2007; Urquhart et al., 2007). Here, we report the unusual genetic behavior of a quantitative gain-of-function *CNGC* mutation (*brush*) in *Lotus japonicus* resulting in a leaky tetrameric channel. *brush* resides in a cluster of redundant *CNGCs* encoding subunits which resemble metazoan voltage-gated potassium (Kv1-Kv4) channels in assembly and gating properties. The recessive mongenic *brush* mutation impaired root development and infection by nitrogen-fixing rhizobia. The *brush* allele exhibited quantitative behavior since overexpression of the cluster subunits was required to suppress the *brush* phenotype. The results reveal a mechanism by which quantitative competition between channel subunits for tetramer assembly can impact the phenotype of the mutation carrier.
DOI: https://doi.org/10.7554/eLife.25012.001

*For correspondence:
parniske@lmu.de

†These authors contributed equally to this work

Competing interests: The authors declare that no competing interests exist.

The legume-rhizobium symbiosis offers an excellent model system to study the role of $Ca^{2+}$ signaling in eukaryotic cell development. Rhizobia produce lipochitooligosaccharides (LCOs) which stimulate signal transduction processes involving not only oscillations of $[Ca^{2+}]$ in the nucleus and perinuclear region but also rapid influx of calcium ions into the cytoplasm of legume root hairs (*Felle et al., 1999*; *Cardenas et al., 1999*; *Ehrhardt et al., 1996*; *Harris et al., 2003*; *Miwa et al., 2006*), preceeding rhizobial entry and organ development. The *Lotus japonicus* mutant *brush* was previously isolated in a screen of an ethyl-methanesulfonate (EMS)-mutagenised population for plants defective in symbiotic cell development (*Maekawa-Yoshikawa et al., 2009*). At 26°C, *brush* roots are stunted and root hair infection threads do not progress into the root cortex, resulting in the formation of non-infected ('empty') nodules. The evidence suggested that the recessive mutation was negatively interfering with infection thread progression and cell expansion in the root apical meristem. The *brush* mutation was mapped to the short arm of chromosome 2 at 8.8 cM (*Maekawa-Yoshikawa et al., 2009*), linked to the marker TM0312. Subsequently, a large-scale recombinant screen for fine-mapping was undertaken. In total, 20 of 1148 tested F2 individuals showed recombination events between the flanking markers TM2432 and TM0348 (*Figure 1—figure supplement 1*). F2 genotyping and subsequent F3 phenotyping refined the target region to 37 kb. One EMS-induced mutation was detected in the first exon of *BRUSH*, a predicted *CNGC* of unknown function. Because the *brush* mutant phenotype could not be complemented with the genomic region containing *BRUSH* including its native promoter (see below), we searched for additional possible missed mutations. The genome of *brush* was sequenced and aligned with the reference genome. Within the already delineated target interval the mutation in *BRUSH* was confirmed and no additional polymorphisms relative to the Gifu wild-type were detected.

**eLife digest** Plants constantly monitor and respond to changes in their environment. Central to this surveillance system is the movement of calcium ions into and out of cells. Calcium ions are normally kept at very low levels inside of cells and subtle changes in these levels relay information about the external environment. In the case of plant roots, changes in the concentration of calcium ions herald essential information about soil conditions and the presence of microorganisms, and in turn trigger appropriate responses.

Calcium ion signals are essential for peas, beans and other members of the legume family to form close relationships (known as symbioses) with soil bacteria called rhizobia. As such, many studies of calcium signalling have focused on root symbioses, particularly in a model legume called *Lotus japonicus*. Previous studies have identified one mutant version of this plant, called *brush*, which develops abnormal roots with brush-like arrays of root hairs near the tip. The *brush* mutant was also unable to form a symbiosis with rhizobia, and structures that allow the bacteria to enter the plant stopped developing before they were complete. However, the gene responsible had not been identified.

Chiasson, Haage et al. set out to identify the responsible mutation. At first the *brush* mutation escaped identification because a key experiment gave an unexpected result. The introduction of a normal, or wild type, copy of the proposed gene – referred to as *BRUSH* – into the *brush* mutant did not correct the problems with its roots. Further analysis revealed that it was actually the ratio between *BRUSH* and *brush* expression levels that was critical for determining how the plant's roots developed.

The mutation in *brush* causes a small change in a protein belonging to the CNGC family. These proteins act as channels and allow ions to move across cell membranes. Further experiments found that the channel formed by the mutated CNGC protein is leaky and allows calcium ions to enter the cell in the absence of any cue from the environment. The leaky entry of calcium ions likely confuses the plant's surveillance system, which disturbs the normal development of the root. It is also likely that the *brush* mutation's effects on calcium signaling also interfere with the entry of rhizobia into the roots. These findings provide important insights into the function of CNGCs and reveal how a small change in a channel protein can have far reaching effects on an organism.

DOI: https://doi.org/10.7554/eLife.25012.002

The predicted genomic sequence of *BRUSH* carried a guanine to adenine (G401A) transition in the first exon in *brush* (*Figure 1A*). Amplification of *brush* cDNA revealed an open-reading frame encoding a protein containing 773 amino acids with an amino acid exchange from glycine to glutamic acid (G134E). The genome of the model plant *Arabidopsis thaliana* contains 20 predicted *CNGC* genes, which can be classified into four distinct groups (I, II, III, IV) (*Mäser et al., 2001*). Phylogenetic and synteny analysis revealed that BRUSH (CNGC.IVA1) is orthologous to the Group IVA members AtCNGC19 and AtCNGC20 (*Figure 1B*). Similar to other Group IVA CNGCs, BRUSH contains a relatively long N-terminal extension followed by six predicted transmembrane domains and a cyclic nucleotide-binding domain (*Figure 1C*). The *brush* mutation is located in a conserved region previously identified as a putative sorting signal in Group IVA CNGCs (*Yuen and Christopher, 2013*). Sequencing coupled with gene prediction of the *brush* target region revealed that *BRUSH* resides in a cluster containing five *CNGC* loci (*Figure 1—figure supplement 2*). Analysis of syntenic genomic regions in legumes and non-legumes revealed that the *CNGC.IVA* cluster expansion occurred early in the legume lineage and was retained. Transcripts for three of the loci could be amplified (*CNGC.IVA3*, *CNGC.IVA4*, *CNGC.IVA5*) and encode proteins which are closely related to BRUSH (*Figure 1—figure supplement 3*). No transcript was detected for *CNGC.IVA2* which contains a large transposon insertion in the seventh intron (*Figure 2A*).

To confirm that *brush* carried the causative mutation, the *BRUSH* genomic sequence was expressed in *brush* transgenic hairy roots driven by its native 2 kb promoter. Surprisingly, we did not observe rescue of either the root or infection thread phenotype (*Figure 2—figure supplement 1A–C*). However, when *brush* was transformed with the *BRUSH* genomic sequence driven by the *L. japonicus* constitutive *polyubiquitin* promoter, we observed restoration of the root and infection thread

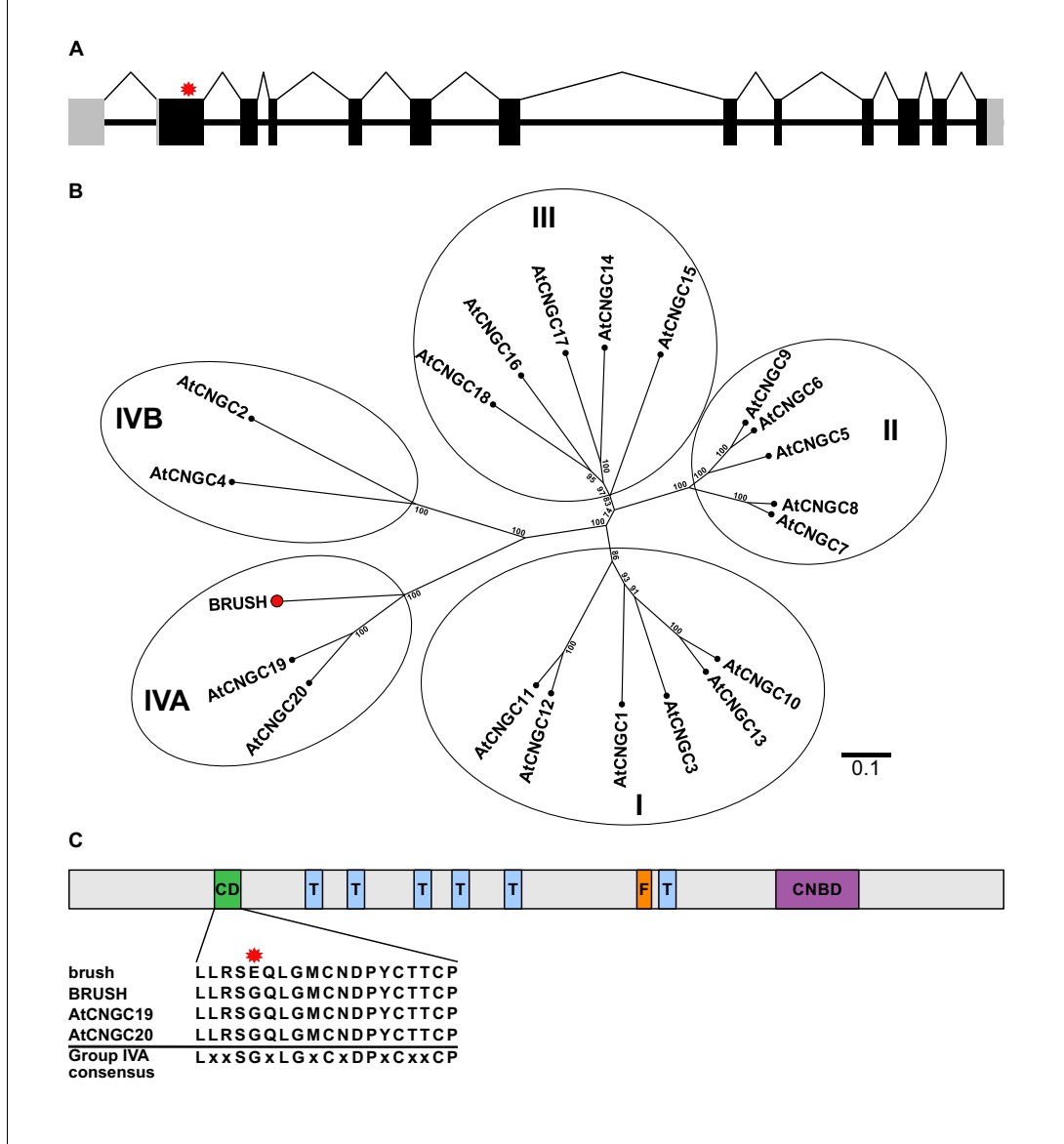

**Figure 1.** *brush* contains a point mutation in *CNGC.IVA1* (*BRUSH*). (**A**) Schematic of the intron-exon structure of *BRUSH* highlighting the *brush* mutation in the first exon (red asterisk). (**B**) Phylogenetic tree of BRUSH (red node end) in relation to *Arabidopsis thaliana* CNGC proteins. BRUSH is orthologous to the Group IVA members AtCNGC19 and AtCNGC20. (**C**) Overview of the BRUSH protein domain structure highlighting the conserved Group IVA domain (CD, green), transmembrane domains (T, light blue), putative filter region (F, orange), and the predicted cyclic nucleotide-binding domain (CNBD, purple). Shown below is the CD sequence in brush (G134E mutation, asterisk) relative to BRUSH, AtCNGC19, and AtCNGC20 and the Group IVA CNGC consensus (*Yuen and Christopher, 2013*). Numbers at the branch points in (**B**) indicate the percentage bootstrap values (100 iterations) for the inferred tree. Scale bar in (**B**) indicates the number of amino acid substitutions per site.

DOI: https://doi.org/10.7554/eLife.25012.003

The following source data and figure supplements are available for figure 1:

**Source data 1.** *Figure 1B* source data.
DOI: https://doi.org/10.7554/eLife.25012.007

**Figure supplement 1.** Map-based cloning of the *brush* mutation on chromosome 2.
DOI: https://doi.org/10.7554/eLife.25012.004

**Figure supplement 2.** Syntenic chromosomal locations of *CNGC.IVA* genes in selected plant species.
DOI: https://doi.org/10.7554/eLife.25012.005

**Figure supplement 3.** Protein sequence comparison of Group IVA CNGCs from *Lotus japonicus*.
DOI: https://doi.org/10.7554/eLife.25012.006

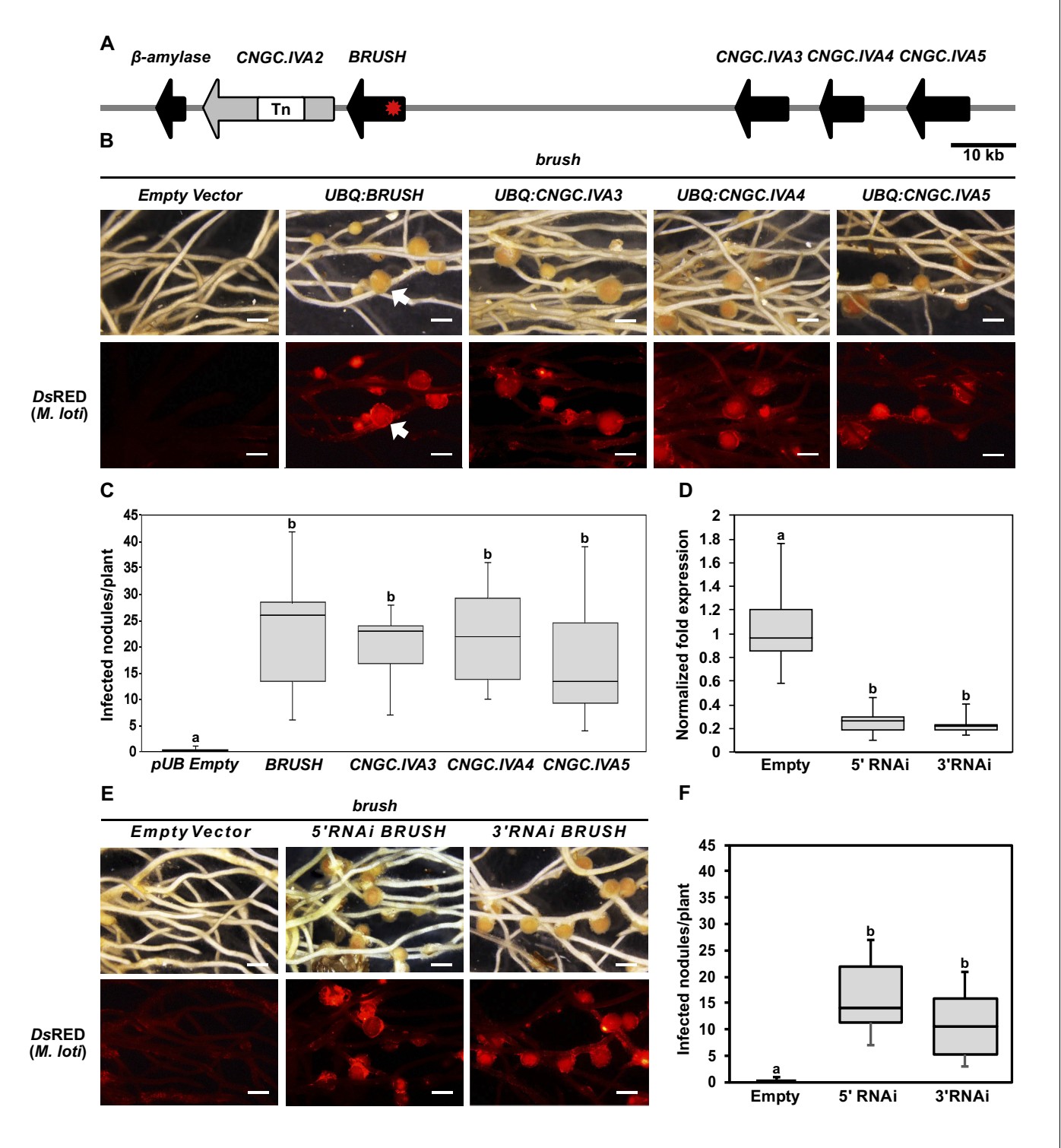

**Figure 2.** Genetic complementation of *brush*. (A) Genomic region surrounding *BRUSH* on *L. japonicus* chromosome 2 showing a cluster of five *CNGC. IVA* genes (red asterisk, *brush* causative mutation). *CNGC.IVA2* contains a large transposon (Tn) insertion. *CNGC.IVA2* contains a large transposon insertion (shown as a gap). (B) Complementation assay of *brush* roots by overexpression (*ubiquitin* promoter, *UBQ*) of the four expressed *CNGC.IVA* cluster members (bright field, top panel). The presence of red fluorescent nodules (arrow) colonized by *Mesorhizobium loti* expressing the red fluorescent protein *Ds*RED (lower panel) indicates successful bacterial infection and thus complementation. (C) Number of nodules per transformed plant from (B) (n = 10 for all constructs). (D) Quantitative reverse-transcription PCR analysis of *brush* transcript levels after RNAi targeting either the 5'UTR or 3' UTR of *brush*. The normalized fold expression of *brush* is shown relative to empty vector control roots (n = 6 for all constructs). (E)

*Figure 2 continued on next page*

*Figure 2 continued*

Complementation analysis of *brush* expressing RNAi fragments targeting either the 5'UTR or 3'UTR of *brush* in the *brush* mutant. Panels are the same as (B). (F) Number of nodules per transformed plant (n = 10 for all constructs) from (E). Roots for both complementation experiments were observed 6 weeks after inoculation with rhizobia. Scale bars in (B) and (E) represent 1 mm. Letters in (C), (D), and (F) indicate different statistical groups (ANOVA followed by Tukey's HSD test). $F_{(4, 45)}$=10.86, p < 0.001 (C), $F_{(2, 15)}$=20.82, p < 0.001 (D), $F_{(2, 27)}$=22.72, p < 0.001 (F).

DOI: https://doi.org/10.7554/eLife.25012.008

The following source data and figure supplements are available for figure 2:

**Source data 1.** *Figure 2C* source data.
DOI: https://doi.org/10.7554/eLife.25012.015
**Source data 2.** *Figure 2D* source data.
DOI: https://doi.org/10.7554/eLife.25012.016
**Source data 3.** *Figure 2F* source data.
DOI: https://doi.org/10.7554/eLife.25012.017
**Source data 4.** *Figure 2—figure supplement 1B* source data.
DOI: https://doi.org/10.7554/eLife.25012.018
**Source data 5.** *Figure 2—figure supplement 3A,C* source data.
DOI: https://doi.org/10.7554/eLife.25012.019
**Source data 6.** *Figure 2—figure supplement 4B* source data.
DOI: https://doi.org/10.7554/eLife.25012.020
**Source data 7.** *Figure 2—figure supplement 5A* source data.
DOI: https://doi.org/10.7554/eLife.25012.021
**Figure supplement 1.** Expression of the *BRUSH* native genomic sequence in *brush*.
DOI: https://doi.org/10.7554/eLife.25012.009
**Figure supplement 2.** Complementation of *brush* by overexpression of *BRUSH*.
DOI: https://doi.org/10.7554/eLife.25012.010
**Figure supplement 3.** RNAi off-target controls and phenotypes associated with either overexpression of *brush* or a null *BRUSH* allele.
DOI: https://doi.org/10.7554/eLife.25012.011
**Figure supplement 4.** Complementation of *brush* by overexpressing *AtCNGC19* or *AtCNGC20*.
DOI: https://doi.org/10.7554/eLife.25012.012
**Figure supplement 5.** *CNGC.IVA gene* expression in *Lotus japonicus* roots after rhizobial inoculation.
DOI: https://doi.org/10.7554/eLife.25012.013
**Figure supplement 6.** *BRUSH* expression in roots during nodulation.
DOI: https://doi.org/10.7554/eLife.25012.014

phenotypes, including infected nodules (*Figure 2B,C* and *Figure 2—figure supplement 2A,B*). These results suggested that expression level of *brush* is critical for phenotype manifestation. To further analyze the relationship between *brush* expression levels and the observed phenotypes, we generated RNA interference (RNAi) constructs to target the untranslated regions (3'UTR or 5'UTR) of the *brush* transcript in the *brush* mutant. Transformation of each RNAi construct specifically silenced *brush* (*Figure 2D*, *Figure 2—figure supplement 3A*) and restored rhizobial infection of root cells (*Figure 2E,F*). We then overexpressed the *brush* allele in wild-type Gifu hairy roots to recapitulate the *brush* phenotype and observed that ectopic overexpression of *brush* impaired hairy root emergence (*Figure 2—figure supplement 3B*). Collectively, these results suggest that the expression level of *brush* is critical for the observed phenotypes and that the phenotypic penetrance of the allele appears to be dosage-dependent. An EMS mutant (SL1484-1) was then obtained by TILLING (*Perry et al., 2009*; *Perry et al., 2003*) containing a point mutation (W119stop) early in the *BRUSH* open-reading frame. Analysis of homozygous mutant plants did not reveal any phenotypic root or infection abnormalities after inoculation with rhizobia (*Figure 2—figure supplement 3C*). The finding that the null mutant of *BRUSH* is not recapitulating the *brush* phenotype indicates that *brush* is an interfering allele and that the loss of *BRUSH* is compensated by potential redundancy of other *CNGC*s within the cluster.

To determine if the other *CNGC.IVA* cluster genes are redundant with respect to *BRUSH*, we overexpressed *CNGC.IVA3*, *CNGC.IVA4*, *CNGC.IVA5* in the *brush* mutant by hairy root transformation. Analysis of transgenic roots revealed that expression of each gene complemented *brush*, as evidenced by colonized root nodules (*Figure 2B,C*). Further, we found that overexpression of the predicted *Arabidopsis* orthologs *AtCNGC19*, or *AtCNGC20* in *brush* also restored nodulation

(*Figure 2—figure supplement 4A,B*). To assess if each *L. japonicus* gene is expressed in roots, RT-qPCR was performed before and after inoculation with rhizobia. Transcripts were detected for *BRUSH, CNGC.IVA3, CNGC.IVA4,* and *CNGC.IVA5*, the levels of which did not show significant changes (<2 fold) after inoculation with the rhizobial symbiont (*Figure 2—figure supplement 5A*). Spatial expression analysis of *promoter:β-glucuronidase* (*GUS*) fusions revealed *BRUSH* expression in root hairs and developing nodules after inoculation with rhizobia and that *CNGC.IVA3, CNGC.IVA4,* and *CNGC.IVA5* are expressed in similar domains (*Figure 2—figure supplement 5B*). Closer investigation of the *BRUSH_{promoter}:GUS* activity revealed a lack of expression in roots prior to inoculation and subsequent activity associated with infected root hairs and nodule primordia after inoculation (*Figure 2—figure supplement 6*). The overlapping expression pattern of the Group IVA *CNGCs* together with their ability to dampen the *brush* phenotype indicate that these genes are redundant.

Given that plant CNGCs are anticipated to form both homomeric and heteromeric tetramers (*Ma and Berkowitz, 2011*), an interaction between BRUSH and redundant CNGCs is conceivable. We initially utilized the yeast split-ubiquitin interaction assay to determine the location of the BRUSH termini (*Xing et al., 2016*). The assay utilizes the N-terminal (Nub) and C-terminal (Cub) fragments of yeast ubiquitin (Ubi4) (*Stagljar et al., 1998*). Reconstitution of ubiquitin in the cytoplasm leads to proteolytic release of Cub (fused to LexA-VP16) and activation of genetic reporters. We observed that BRUSH-Cub interacted with both NubI-BRUSH and BRUSH-NubI fusions demonstrating that both BRUSH termini are located in the cytoplasm (*Figure 3—figure supplement 1*). Since voltage-gated ion channel subunits interact via their soluble domains (*Barros et al., 2012*), we focused on the CNGC.IVA soluble termini for yeast two-hybrid interaction assays. We observed a self-interaction for the BRUSH N-terminus (NT) as well as interaction with the NTs of brush, CNGC.IVA3, CNGC.IVA4, CNGC.IVA5 (*Figure 3A*) along with AtCNGC19 and AtCNGC20 (*Figure 2—figure supplement 4C*). To further substantiate the interaction observed in yeast, we co-injected full-length subunits into *Xenopus laevis* oocytes for bimolecular fluorescence complementation (BiFC) assays. Expression of BRUSH-BRUSH, BRUSH-brush, and brush-brush combinations resulted in successful complementation (*Figure 3—figure supplement 2A,B*). The yeast and oocyte interaction assays demonstrate that CNGC.IVA channels potentially form homo- and hetero-complexes in vivo, which may be mediated in part by their NT domains.

To characterize their channel properties, we injected either *BRUSH* or *brush* into *Xenopus* oocytes for two-electrode voltage clamping. Expression was confirmed for both BRUSH-YFP and brush-YFP by confocal microscopy (*Figure 3B,C*). Oocytes expressing BRUSH (*Figure 3B,D*) or BRUSH-YFP (*Figure 3—figure supplement 2C,E*) failed to yield significant inward currents at negative voltages in the presence of up to 30 mM CaCl$_2$. In contrast, under the same experimental conditions, oocytes expressing brush (*Figure 3C,D*) or brush-YFP (*Figure 3—figure supplement 2D,E*) produced voltage- and time-dependent inward currents at negative voltages. The currents were evident starting from 15 mM external CaCl$_2$ and increased in a dose-dependent manner with the external CaCl$_2$ concentration (*Figure 3E*). Exchange of Ca$^{2+}$ as a charge carrier to K$^+$ abolished the voltage-dependent inward currents mediated by brush-YFP (*Figure 3—figure supplement 2D,F*), indicative of a hyperpolarization-activated Ca$^{2+}$-permeable channel.

Since an N-terminal missense mutation leads to activation of brush, the conserved CNGC.IVA cytoplasmic domain may mediate channel gating (*Figure 4A*). The expression of *brush* alone in oocytes induces Ca$^{2+}$ influx, therefore assembly of a brush homocomplex leads to deregulated activity (*Figure 4B*). As *brush* is recessive we speculate that brush is mainly positioned in silent heterotetrameric complexes in the heterozygous state, but assembles into a population of homocomplexes in homozygous plants triggering the phenotype (*Figure 4C*). Given that *brush* is expressed in root hairs and nodule primordia after inoculation with rhizobia and that Ca$^{2+}$ spiking in *brush* is intact (*Maekawa-Yoshikawa et al., 2009*), the deregulated Ca$^{2+}$ influx activity may impair rhizobial infection progression by interfering with downstream signaling events.

*brush* is a rare recessive gain-of-function missense mutation and exhibits an unusual quantitative genetic behavior. We pinpointed the CNGC tetrameric complex in combination with the expanded gene family as the causative factors for the unusual genetics. Although definitive evidence demonstrating that plant CNGCs form tetramers is required, their inclusion in the superfamily of voltage-gated ion channels predicts they will form such complexes. In support of this conclusion, the recent cryo-electron microscopy structures of the TAX-4 CNG channel from *Caenorhabditis elegans* (*Li et al., 2017*), the prokaryotic LliK CNG channel from *Leptospira licerasiae* (*James et al., 2017*),

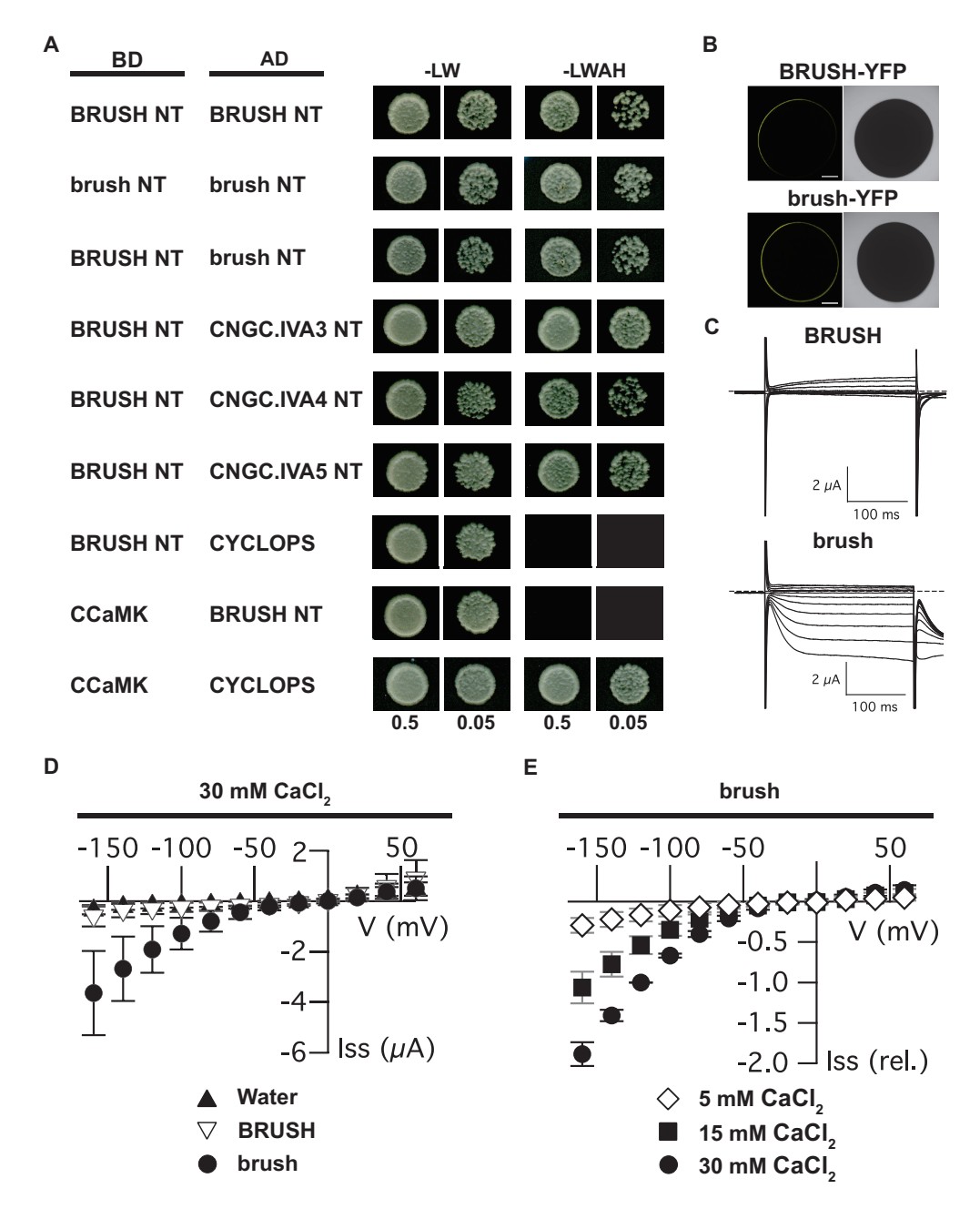

**Figure 3.** Interaction of CNGC.IVA N-termini in yeast and channel activity in *Xenopus* oocytes. (A) Yeast two-hybrid interaction of the soluble BRUSH N-terminus (NT) fused to the GAL4-binding domain (BD) and the NT of the indicated CNGC.IVA proteins fused to the GAL4 activation domain (AD). Yeast cells were resuspended in water (OD$_{600}$= 0.5 and 0.05) and spotted onto -LW (leucine, tryptophan) and -LWAH (leucine, tryptophan, adenine, histidine) solid media. (B) Confocal fluorescence images of oocytes expressing either BRUSH-YFP or brush-YFP fusion proteins. (C) Plasma membrane currents of oocytes expressing BRUSH or brush in the presence of 30 mM CaCl$_2$. Voltage steps ranged from +60 to −160 mV in 20 mV increments, starting from a holding potential of −40 mV. Dashed lines indicate 0 µA. (D) Current-voltage relations of oocytes injected with water or YFP (▲, n=15), BRUSH or BRUSH-YFP (▽, n=17), and brush or brush-YFP (•, n=26) . (E) Relative current-voltage relations for oocytes expressing brush and brush-YFP in the presence of 5 mM CaCl$_2$ (◇ , n = 3), 15 mM CaCl$_2$ (■, n = 3), and 30 mM CaCl$_2$ (•, n = 6). Currents are shown relative to the current at −120 mV in the presence of 30 mM CaCl$_2$. Data in (D) and (E) represent mean values ± standard deviations. Scale bars in images (B) represent 250 µm.
DOI: https://doi.org/10.7554/eLife.25012.022

The following source data and figure supplements are available for figure 3:

**Source data 1.** *Figure 3D* source data.
DOI: https://doi.org/10.7554/eLife.25012.025

*Figure 3 continued*

**Source data 2.** *Figure 3E* source data.
DOI: https://doi.org/10.7554/eLife.25012.026
**Source data 3.** *Figure 3—figure supplement 2E* source data.
DOI: https://doi.org/10.7554/eLife.25012.027
**Source data 4.** *Figure 3—figure supplement 2F* source data.
DOI: https://doi.org/10.7554/eLife.25012.028
**Figure supplement 1.** BRUSH topology determination in yeast.
DOI: https://doi.org/10.7554/eLife.25012.023
**Figure supplement 2.** Brush interaction in oocytes and permeability to $Ca^{2+}$ and $K^+$.
DOI: https://doi.org/10.7554/eLife.25012.024

and the human hyperpolarization-activated cyclic nucleotide-gated (HCN1) channel (*Lee and MacKinnon, 2017*) all disclose a tetrameric assembly. Therefore, we speculate that quantitative competition amongst redundant subunits for tetramer inclusion clarifies why a *BRUSH* null is phenotypically wild type and why overexpression is required to suppress the *brush* phenotype.

Expression of brush in oocytes revealed that the mutation renders the channel permeable to $Ca^{2+}$ influx under hyperpolarizing conditions. Evidence obtained from Arabidopsis (*Gao et al., 2016*; *Wang et al., 2017*; *Zhang et al., 2017*), *Medicago truncatula* (*Charpentier et al., 2016*), and the moss *Physcomitrella patens* (*Finka et al., 2012*) CNGCs also supports the inward rectification of $Ca^{2+}$ by plant CNGCs, while numerous physiological studies have implicated CNGCs as being intimately linked to $Ca^{2+}$ (*Wang et al., 2013*; *Guo et al., 2010*; *Chan et al., 2003*; *Urquhart et al., 2007*). Although brush was impermeable to $K^+$ in our assay, evidence exists that some plant CNGCs are permeable to other cations ($K^+$ and $Na^+$) in heterologous systems (*Gobert et al., 2006*; *Ali et al., 2006*; *Leng et al., 2002*; *Leng et al., 1999*). Collectively, the evidence demonstrates that plant CNGCs inwardly rectify cations.

Our results demonstrate that plant CNGC.IVAs may share more in common with metazoan Kv1-Kv4 $K^+$ channels relative to typical mammalian CNGs. Similar to BRUSH, Kv1-Kv4 channel gating and subunit interactions are mediated by an N-terminal T1 domain (*Barros et al., 2012*; *Zagotta et al., 1990*). In contrast, human CNGs assemble via C-terminal interactions and are gated by binding of cyclic nucleotides (*Barros et al., 2012*; *Giorgetti et al., 2005*). In addition to CNGCs, plants contain a family of shaker-type $K^+$ channels with cyclic nucleotide-binding domains. Similar to BRUSH, these channels are not gated by cyclic nucleotides, but instead are regulated by voltage and relative ion concentrations (*Hedrich, 2012*). Since plant CNGCs have been difficult to assess in heterologous systems (*Ma and Berkowitz, 2011*), the discovery that a single residue substitution in a conserved domain is sufficient for activation represents a significant advance towards understanding their regulation.

## Materials and methods

### Plant material and transformations

*Lotus japonicus* Gifu (wild-type, accession B-129) (*Handberg and Stougaard, 1992*), Miyakojima (accession MG-20) (*Kawaguchi et al., 2001*) and *brush* (EMS mutant SL0979-2, Gifu) (*Perry et al., 2003*) plants were used. The *BRUSH* TILLING line SL1484-1 was obtained from the *L. japonicus* TILLING facility (John Innes Centre, Norwich, UK). The seed bag numbers of critical lines are listed in *Supplementary file 3*. Seeds were scarified with sandpaper, sterilized for 10 min in 4% sodium hypochlorite, and imbibed overnight in sterile water at 4°C. Hairy roots were generated using the *Agrobacterium rhizogenes* strain AR1193 (*Stougaard et al., 1987*). Nodulation experiments were carried out by inoculating plants grown in pots or weck jars containing a sand-vermiculite mixture and Fåhraeus (*Fahraeus, 1957*) media with *Mesorhizobium loti* MAFF303099 expressing DsRed (*Markmann et al., 2008*). Transgenic roots were visualized with either a stereomicroscope (Leica M165FC) or confocal laser scanning microscope (Leica SP5). Hairy roots were stained for GUS and sectioned as described previously (*Chiasson et al., 2014*). Plants were cultivated in growth cabinets at 22°C (16 hr light/8 hr dark). All complementation and GUS experiments were carried out a

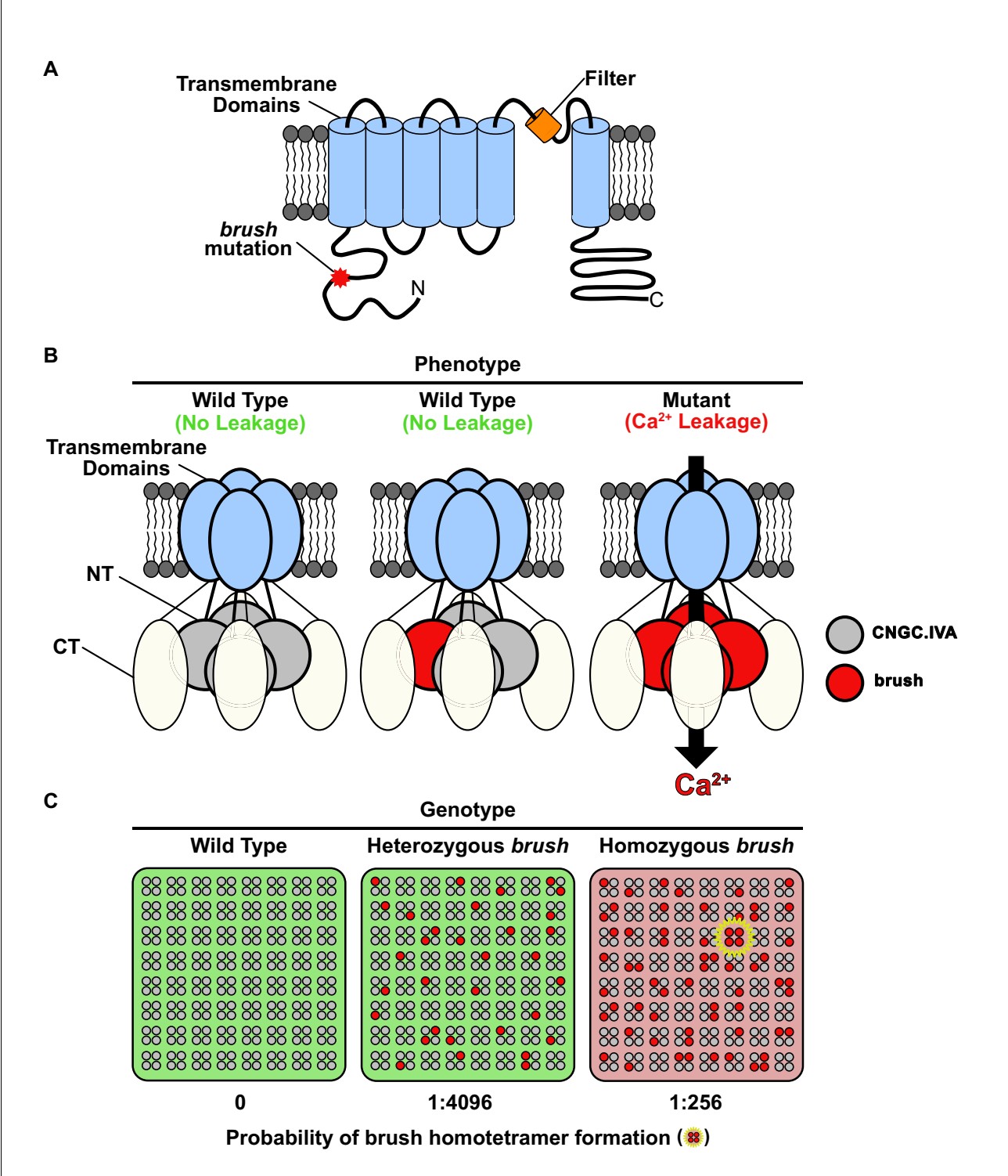

**Figure 4.** Model for *brush* activity *in planta*. (A) The predicted two-dimensional topology of a brush monomer embedded in a lipid bilayer. (B) Model explaining the mechanistic connection between the relative abundance of the brush mutant channel with the unusual quantitative genetic behavior of the *brush* phenotype. Based on complementation and interaction studies, BRUSH engages in a tetrameric complex along with 3 other CNGC.IVA proteins. From the segregation analysis, plants heterozygous for *brush* exhibit a wild-type phenotype. Expression of *brush* in oocytes mediates leaky $Ca^{2+}$ influx, indicating that a homotetramer is inappropriately active. (C) Probability-based overview of brush homotetramer formation assuming that each of the 4 CNGC.IVA subunits participates with equal likelihood in complex formation. Shown is a grid containing 8 × 8 tetramers with CNGC. IVA WT (grey dots) and brush (red dots) subunits. Both wild-type and heterozygous plants do not exhibit a phenotype (indicated by green background). *Figure 4 continued on next page*

*Figure 4 continued*

The probability of forming brush homotetramer is 1:4096 for a heterozygous and 1:256 for a *brush* homozygous genotype. A brush homotetramer (shown inside yellow star) is required to trigger the phenotype (red background). This frequency-dependent assembly of the leaky brush tetramer explains the phenotypic differences observed in plants harboring different allele frequencies.
DOI: https://doi.org/10.7554/eLife.25012.029

minimum of three times and displayed similar results. Crossings were performed as described previously (*Jiang and Gresshoff, 1997*). Primers and plasmids used for all experiments are listed in *Supplementary file 1* and *Supplementary file 2*, respectively.

## Map-based cloning of the *brush* mutation

F2 plants from a cross between *brush* and MG-20 were used for fine mapping using SSR markers as described (*Groth et al., 2013*). Primer sequences were obtained from the Kazusa DNA Research Institute website (http://www.kazusa.or.jp/lotus/markerdb_index.html). The region was further refined using identified SNPs. The *brush* target interval between TM2432 and SNP3 (approximately 103 kb) was sequenced by Sanger sequencing. The *brush* genome was also reassembled after next-generation sequencing to identify mutant-specific polymorphisms. Nuclear DNA (see below) of *brush* seedlings was subjected to next-generation sequencing at Eurofins MWG, Germany, using an Illumina HiSeq 2000 (Illumina, USA) with a read length of $2 \times 100$ bp. Genes in the *brush* target region were annotated after sequencing using Genscan (*Burge and Karlin, 1997*) and Artemis (*Rutherford et al., 2000*). CLC Genomics Workbench (CLC bio, Denmark) was used to analyze the sequencing data.

## Nuclear DNA extraction for next-generation sequencing

Four-week-old *brush* seedlings were transferred to the dark for 2 days before leaf material was harvested. Approximately 2 g of ground powder was resuspended in 20 ml ice-cold HB buffer (10 mM Tris, 80 mM KCl, 10 mM EDTA, 1 mM spermine, 1 mM spermidine, 0.5 M sucrose, 0.5% triton X-100, 0.15% β-mercaptoethanol, pH 9.4 with NaOH) by gentle shaking on ice. The solution was filtered through two layers of Miracloth (Calbiochem, Merck, Germany). The flow-through was transferred to a 15-ml Falcon tube and the nuclei were pelleted at 4°C by centrifugation (1800 x g) and washed two times by resuspension in HB buffer. The final pellet was resuspended in 500 µl CTAB buffer (55 mM cetyltrimethylammonium bromide, 1.4 M NaCl, 20 mM EDTA, 100 mM Tris, pH 8), and incubated at 60°C for 30 min. 500 µl chloroform:isoamylalcohol (24:1) was added and mixed by inverting the tube several times. After a centrifugation step at 8000 x g (4°C) for 10 min, the upper phase was transferred to a new tube. 5 µl of RNase (10 mg/ml stock concentration) was added and incubated at 37°C for 30 min. 0.6 volumes ice-cold isopropanol was added and mixed by inverting the tube several times. The nuclear DNA was then precipitated at −20°C overnight and centrifuged for 10 min at 16,000 x g and 4°C. The supernatant was discarded and the pellet was washed with 70% ethanol and resuspended in 55 µl TE buffer.

## Yeast two-hybrid and split-ubiquitin assays

Yeast two-hybrid interaction assays were conducted with the haploid yeast strain AH109 (Clontech). Split ubiquitin interaction assays were carried out in the haploid strain THY.AP4 (*Obrdlik et al., 2004*). THY.AP4 and plasmids for split-ubiquitin were obtained from the Arabidopsis Biological Resource Center (http://abrc.osu.edu/). Plasmids used for both interaction assays are shown in *Supplementary file 2*. Bait and prey plasmids were introduced via double transformation using the lithium acetate method (*Gietz and Schiestl, 2007*) and selected on media lacking leucine and tryptophan (-LW). The interacting protein pair of CCaMK and CYCLOPS was used as a control for yeast two-hybrid (*Yano et al., 2008*). Positive transformants were restreaked on -LW, then used to inoculate overnight cultures in liquid -LW media. Overnight cultures were diluted to $OD_{600}$ of 0.5 in sterile water and diluted 10-fold. 5 µl was spotted on –LW or solid media lacking leucine, tryptophan, adenine, and histidine (-LWAH). Yeast plates were incubated at 28°C for 3–5 days. All interaction assays were independently conducted a minimum of three times.

## Clone preparation for *Xenopus* oocyte experiments

*BRUSH* and *brush* coding sequences were cloned for *Xenopus* expression with a custom Golden Gate cloning strategy using a modified backbone obtained from the Standard European Vector Architecture 2.0 database (*Martinez-Garcia et al., 2015*). The backbone (with flanking bacterial transcriptional terminators) was derived from pSEVA191 (http://wwwuser.cnb.csic.es/~seva/) and was chosen to alleviate toxicity issues uncovered while cloning *CNGC.IVA* sequences into pUC-based Golden Gate backbones and pGEMHE (*Liman et al., 1992*). A *ccdB* cassette compatible with Golden Gate cloning (*Binder et al., 2014*) was amplified and inserted into the AvrII/SacI sites of pSEVA191 to create the LII backbone pSEVA191 1–2. The coding sequences of *BRUSH* and *brush* were then combined in a BsaI cut-ligation with modules containing the T7 promoter as well as the 5'UTR and 3'UTR sequences of *β-globin* mRNA (amplified from pEMHE). The same backbone was used to express the constructs for BiFC analysis, where LI Golden Gate B-C or D-E parts encoding for the N-terminal (VN) or C-terminal (VC) portions of mVenus (*Offenborn et al., 2015*) were inserted. Plasmids were assembled in a 15 µl reaction containing 100 ng of each LI plasmid and backbone, 1.5 µl CutSmart buffer (NEB, Germany), 1.5 µl 10 mM ATP, 0.75 µl BsaI (NEB), 0.75 µl T4 ligase (NEB). The reaction was then cycled 6 times (10 min at 37°C, 10 min 16°C) in a PCR machine, followed by incubation at 37°C (10 min) and 65°C (20 min).

## Functional analysis in *Xenopus laevis* oocytes

Capped RNA (cRNA) synthesis, oocyte injection, and voltage-clamp recordings were performed as described (*Becker et al., 2004*; *Müller-Röber et al., 1995*). cRNA was synthesized with a mMESSAGE mMACHINE T7 Transcription Kit (ThermoFisher, Germany) and oocytes were injected (General Valve Picospritzer III, Parker Hannifin Corp.) with approximately 25 ng cRNA or with RNase-free water as a control. Injected oocytes were stored at 18°C in ND96 solution (96 mM NaCl, 2 mM KCl, 1 mM $CaCl_2$, 1 mM $MgCl_2$, 5 mM HEPES, 10 mM sorbitol, pH 7.4 with NaOH) adjusted to 220 mOsm/L with sorbitol and supplemented with 25 µg/ml gentamycin until use. Measurements were recorded 2 to 3 days after injection using the two-electrode voltage-clamp technique with a Turbo Tec-10Cx amplifier (NPI electronic GmbH). During two-electrode voltage clamp measurements, oocytes were constantly perfused with bath solution composed of 30 mM CaCl2, 10 mM MES-Tris pH 7.4, adjusted to 220 mOsm/L with mannitol and supplemented with either 100 µM 8-Bromo-cAMP (Sigma) or 100 µM 8-CPT-cAMP (BioLog). For analysis of channel permeabilities, $CaCl_2$ was exchanged as indicated in the figure legends with 5 mM $CaCl_2$, 15 mM $CaCl_2$, or 60 mM KCl. Starting from a holding potential of −40 mV, voltage steps from +60 to −160 mV in 20 mV increments were applied (PatchMaster, HEKA Electronics Inc.). For localization, YFP was fused to the C-terminus of *BRUSH* or *brush*. Oocytes were imaged by confocal microscopy 2 to 3 days after injection with *BRUSH*-YFP and *brush*-YFP cRNA (Leica TCS SP5, excitation: 488 nm, detection: 525–575 nm) to confirm expression. The same protocol was used for BiFC experiments, except that cRNAs were mixed 1:1 prior to injection.

## Gene expression analysis

For analysis of gene expression after rhizobial inoculation, *Lotus japonicus* Gifu seeds were germinated and grown on half-strength B5 agar plates for 14 days. Six plants were planted per weck jar containing sand/vermiculite with Fåhraeus media. After 7 days, root tissue from a single jar was collected and pooled (represents a biological replicate) for the Day 0 time point. *Mesorhizobium loti* MAFF303099 expressing *Ds*Red was added to the remaining jars and tissue was collected in the same manner after 12 days. To analyze gene expression after RNAi, positive hairy roots were isolated from individual plants 6 weeks after inoculation with *Mesorhizobium loti* MAFF303099 *Ds*Red. For both experiments, root tissue was ground in liquid nitrogen and RNA was extracted with a Spectrum Plant Total RNA Kit (Sigma). Genomic DNA was removed using a Turbo DNA-free Kit (Ambion) and total RNA (1 µg for the time course and 200 ng for RNAi) was used for cDNA synthesis with Superscript III (ThermoFisher). cDNA was then checked for genomic DNA contamination by PCR. Expression of *CNGC.IVA* cluster genes after rhizobia inoculation was analyzed by qPCR using SYBR Select Master Mix (Applied Biosystems) with a CFX96 real-time PCR machine. *brush* expression after RNAi was analyzed by qPCR using mi-real-time EvaGreen Master Mix (Metabion) with a QuantStudio 5 Real-Time PCR System (ThermoFisher). In both cases, the plotted data point for each biological

replicate represents the mean of three technical replicates. The relative expression was calculated with the $2^{-\Delta\Delta CT}$ method (*Schmittgen and Livak, 2008*) using eEF-1Aα (GenBank: BP045727) as the reference.

### Bioinformatics and statistics

*Arabidopsis thaliana* protein sequences were obtained from The Arabidopsis Information Resource (TAIR). A multiple sequence alignment was generated using MUSCLE in CLC Main Workbench (CLC bio, Denmark). A Maximum Likelihood phylogenetic tree was calculated using UPGMA (100 bootstrap iterations were performed). One-way ANOVA statistical analysis of data followed by a post-hoc Tukey's multiple comparisons test and t-tests were calculated using GraphPad Prism.

## Acknowledgements

This work was supported by funding from European Research Council (MP; Project No:340904), Deutsche Forschungsgemeinschaft (MP, PD; Forschergruppe 964), and the Alexander von Humboldt Foundation (DC). We would like to thank C Korbmacher (FAU Erlangen) for providing *Xenopus laevis* oocytes, Gudrun Steingräber (FAU Erlangen) for cRNA synthesis, and Christopher Grefen (University of Tübingen) for advice regarding the split-ubiquitin system.

## Additional information

### Funding

| Funder | Grant reference number | Author |
|---|---|---|
| Deutsche Forschungsgemeinschaft | FOR964 (Calcium) | Kristina Haage Petra Dietrich Martin Parniske |
| FP7 Ideas: European Research Council | 340904 (EvolvingNodules) | Martin Parniske |
| Alexander von Humboldt-Stiftung | | David M Chiasson |

The funders had no role in study design, data collection and interpretation, or the decision to submit the work for publication.

### Author contributions

David M Chiasson, Conceptualization, Supervision, Funding acquisition, Investigation, Writing—original draft, Acquisition of Humboldt Fellowship, Supervision of Katharina Sollweck, Cloning, Gene expression analysis, Yeast interaction assays, Complementation, Promoter analysis, Microscopy; Kristina Haage, Validation, Investigation, Writing—review and editing, Map-based cloning of brush, Complementation; Katharina Sollweck, Investigation, Writing—review and editing, Yeast-two hybrid, Electrophysiology, Complementation and promoter analysis; Andreas Brachmann, Resources, Formal analysis, Writing—review and editing, Sequencing and analysis of the CNGC cluster; Petra Dietrich, Resources, Formal analysis, Supervision, Investigation, Writing—review and editing, Supervision of electrophysiology experiments, Equipment, Conducting electrophysiology and BiFC experiments; Martin Parniske, Conceptualization, Resources, Formal analysis, Supervision, Funding acquisition, Investigation, Project administration, Writing—review and editing, Supervision of doctoral student Kristina Haage

### Author ORCIDs

David M Chiasson · http://orcid.org/0000-0002-0770-2684
Andreas Brachmann · http://orcid.org/0000-0001-7980-8173
Petra Dietrich · https://orcid.org/0000-0002-9209-8089
Martin Parniske · http://orcid.org/0000-0001-8561-747X

### Decision letter and Author response

Decision letter https://doi.org/10.7554/eLife.25012.035
Author response https://doi.org/10.7554/eLife.25012.036

## Additional files

### Supplementary files

• Supplementary file 1. Oligonucleotides used in this study.

DOI: https://doi.org/10.7554/eLife.25012.030

• Supplementary file 2. DNA plasmids used in this study.

DOI: https://doi.org/10.7554/eLife.25012.031

• Supplementary file 3. List of key plant material used in this study.

DOI: https://doi.org/10.7554/eLife.25012.032

• Transparent reporting form

DOI: https://doi.org/10.7554/eLife.25012.033

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
