## [Decision Letter]

Thank you for submitting your article "A quantitative hypermorphic CNGC allele confers ectopic calcium flux and impairs cellular development" for consideration by *eLife*. Your article has been favorably evaluated by Christian Hardtke (Senior Editor) and three reviewers, one of whom is a member of our Board of Reviewing Editors. The reviewers have opted to remain anonymous.

The reviewers have discussed the reviews with one another and the Reviewing Editor has drafted this decision to help you prepare a revised submission.

The manuscript provides new insight into CNGC channels in plants through analysis of an unusual hypomorphic allele of BRUSH, a CNGC channel from *L. japonicus* that is required for nodulation. The strength of the work is the elegant resolution of the genetic riddle of this hypermorphic allele and the identification of a missense mutation in a conserved N-terminal region of the protein which appears to result in a constitutively-open channel. The data are explained by a model involving a tetrameric complex of Brush monomers and suggests assembly and gating properties more similar to those of voltage-gated K^+^ channels rather than the classical animal CNGC channels.

There is still much that is unknown about CNGC channels in plants and the reviewers agree that the discovery of a residue that results in constitutive activation represents a significant advance towards understanding their regulation. However, they feel that two aspects of the data need to be further substantiated.

Protein:Protein interaction data:

The protein interaction data need to be strengthened. As noted below, the yeast 2 hybrid experiments are missing some controls and furthermore, the interactions are based only on the yeast 2 hybrid data. The yeast 2 hybrid data need to be improved and the subunit interactions need to be confirmed in a second system. For example, can BRUSH and brush interactions be observed in oocytes (by FRET or other protein-protein interaction approaches). Does Brush run as a multimeric complex on gels?

Specific concerns about the yeast 2 hybrid data.

The current yeast 2 hybrid data are missing some negative and positive controls. The empty vector control indicates that there is no self-activation but there are no controls with proteins that do not bind (something is needed to demonstrate specificity of binding i.e. another negative control protein that does not bind to the CNGCs). Additionally, does brush interact with itself? This is not shown in the current data and needs to be demonstrated as it is essential for the model.

Calcium measurements to support the model:

It is proposed that a deregulated calcium influx impairing infection is suggested to explain the brush nodulation phenotype. There are robust calcium sensors that could be used to document this effect directly. Comparative measurements of calcium influx in root hairs of brush mutants and wild type root hairs are needed to provide in planta evidence to support the proposed model.

---

## [Author Response]

There is still much that is unknown about CNGC channels in plants and the reviewers agree that the discovery of a residue that results in constitutive activation represents a significant advance towards understanding their regulation. However, they feel that two aspects of the data need to be further substantiated.Protein:Protein interaction data:The protein interaction data need to be strengthened. As noted below, the yeast 2 hybrid experiments are missing some controls and furthermore, the interactions are based only on the yeast 2 hybrid data. The yeast 2 hybrid data need to be improved and the subunit interactions need to be confirmed in a second system. For example, can BRUSH and brush interactions be observed in oocytes (by FRET or other protein-protein interaction approaches). Does Brush run as a multimeric complex on gels?

To confirm the subunit interaction in a second system we conducted BiFC experiments in oocytes and observed that BRUSH-BRUSH, BRUSH-brush, and brush-brush associate in this assay. This data has been added in Figure 3—figure supplement 2. We also attempted purification assays in plants and yeast (both full-length and isolated N-terminus) but encountered complications due to aggregation after overexpression. We were unable to purify sufficient quantities for analysis by gel electrophoresis or co-IPs. We therefore chose to use oocytes as the expression system yielded sufficient amounts of protein to observe the BiFC interaction. The following text was added regarding the BiFC experiment:

“To further substantiate the yeast interaction, we co-injected full-length subunits into *Xenopus laevis* oocytes for bimolecular fluorescence complementation (BiFC) assays. Expression of BRUSH-BRUSH, BRUSH-brush, and brush-brush combinations resulted in successful complementation (Figure 3—figure supplement 2).”

Specific concerns about the yeast 2 hybrid data.The current yeast 2 hybrid data are missing some negative and positive controls. The empty vector control indicates that there is no self-activation but there are no controls with proteins that do not bind (something is needed to demonstrate specificity of binding i.e. another negative control protein that does not bind to the CNGCs). Additionally, does brush interact with itself? This is not shown in the current data and needs to be demonstrated as it is essential for the model.

We have addressed the concerns regarding the yeast two-hybrid controls by including CCaMK and CYCLOPS from *Lotus japonicus*. CCaMK and CYCLOPS interacted in the yeast two-hybrid system while neither protein interacted with the N-terminal domain of the tested CNGCs. The results have been added to Figure 3. Yes, the N-terminus of the brush mutant does maintain the self-interaction and this result has also been added to Figure 3.

Calcium measurements to support the model:It is proposed that a deregulated calcium influx impairing infection is suggested to explain the brush nodulation phenotype. There are robust calcium sensors that could be used to document this effect directly. Comparative measurements of calcium influx in root hairs of brush mutants and wild type root hairs are needed to provide in planta evidence to support the proposed model.

The main focus of the manuscript is the unusual genetics displayed by the mutation. We have generated a proposed model to explain our observations regarding the allele, however the root hair is not currently featured in the model. The original manuscript describing the *brush* mutant phenotype (Maekawa-Yoshikawa et al., 2009) demonstrated that root hair calcium spiking is normal in response to applied Nod factor. Since we have only observed *BRUSH_promoter_:GUS* activity in isolated root hairs after the rhizobial infection process has been initiated, *brush* likely acts downstream of the initial signaling events. Further, as *brush* is not constitutively expressed we don’t believe the activity of mutant channel can be properly assessed in root hairs. We have modified the original text to read:

“Given that *brush* is expressed in root hairs and nodule primordia after inoculation with rhizobia and that Ca^2+^ spiking in *brush* is intact*^10^*, the deregulated Ca^2+^ influx activity may impair rhizobial infection progression by interfering with downstream signaling events.”